# TinyStories: How Small Can Language Models Be and Still Speak Coherent English?

## Abstract

In this work, we introduce **TinyStories**, a synthetic dataset of short stories that only contain words that a typical 3 to 4-year-olds usually understand, generated by GPT-3.5 and GPT-4. We show that TinyStories can be used to train and evaluate LMs that are much smaller than the state-of-the-art models (**below 10 million total parameters**), or have much simpler architectures (**with only one transformer block**), yet still produce fluent and consistent stories with several paragraphs that are diverse and have almost perfect grammar, and demonstrate reasoning capabilities.

We also introduce a new paradigm for the evaluation of language models: We suggest a framework which uses GPT-4 to grade the content generated by these models as if those were stories written by students and graded by a (human) teacher. This overcomes the flaws of standard benchmarks which often require the model's output to be very structured, and moreover provides a multidimensional score for the model, providing scores for different types of capabilities such as grammar, creativity and instruction-following.

## 1 Introduction

Natural language is rich and diverse, requiring not only grammatical and lexical knowledge, but also factual information and contextual reasoning. How well can language models (LMs) (4; 5; 18) generate coherent and fluent text, and what are the minimal requirements for this ability? So far, the evidence suggests that small language models (SLMs) with around 125M parameters (3; 20) or less struggle to produce any consistent text beyond a few words, even after extensive training on large and diverse corpora. This raises the question of whether the emergence of the ability to generate coherent English text requires large models and complex architectures, or whether this difficulty is a result of the immense diversity and the amount of details contained in the language corpora that these models are trained on (11; 27; 1). This leads to the question: Is it possible to design a dataset that preserves the essence of natural language, while reducing its breadth and diversity?

In this paper, we introduce TinyStories, a synthetic dataset of short stories that only contain words that a typical 3 to 4-year-olds usually understand, generated by GPT-3.5 and GPT-4. TinyStories is designed to capture the core elements of natural language, such as grammar, vocabulary, facts, and reasoning, while spanning the knowledge base of a 3-4 year old child (7; 17). Based on this dataset, our paper makes several main contributions:

- We show that TinyStories can be used to train and evaluate SLMs that are much smaller than the state-of-the-art models (below 10 million parameters), or have much simpler architectures (with only one transformer block), yet still produce a diverse set of fluent and consistent stories that are comparable or superior to those generated by larger and more complex models. Moreover, we show that these models exhibit emergent reasoning capabilities, knowledge of general facts and ability to follow certain instructions.

- We introduce a new paradigm for evaluating language models using GPT-4, which overcomes many of the limitations of standard benchmarks.

- We demonstrate that despite that fact that training generative models on TinyStories can be done in less than a day on a single GPU, they still exhibits similar behaviors to those observed

in large language models (LLMs), such as scaling laws, trade-offs between width and depth, etc. We conduct extensive experiments to study the effects of different hyperparameters and architectures on the performance and quality of the models.

- We show that the trained SLMs are substantially more interpretable than larger ones. We visualize and analyze the attention and activation maps of the models, and show how they relate to the generation process and the story content. Due to space limit, this can be found in Appendix E.

To give the reader a first impression of the abilities of models, trained on TinyStories, a comparison between completion of a 28M parameter model trained on TinyStories[1] with that of GPT2-XL, which is two orders of magnitude bigger (1.5B parameters), on sample prompt[2], is given in Figure 1. Arguably, the completion provided by our 28M-parameter model is of higher quality than the one provided by GPT2-XL. We remark that the architectures and training scheme of the models are essentially the same.

| **Prompt** | Tom and Jane are friends. One day, Jane goes to Tom's house. Tom has a big pot of soup. He wants to share it with Jane. "Jane, do you want some soup?" Tom asks. "Yes, please. It looks yummy," Jane says. Tom pours some soup into two bowls. He gives one bowl to Jane. Jane takes a spoonful of soup, but then she makes a face. The soup is |
|---|---|
| Completion by GPT2-XL (**1.5B** parameters) completion | too old. Tom scowls.
"Don't be that way! Your soup has to be fresh like your daughter," Tom says, angry and disappointed. "Bring me some more," Jane cries. "No! Go away, I like my soup old and fresh!" |
| Completion by a **28M**-parameters model trained on TinyStories | very bitter. She does not like it. She says, "I don't like this soup. It is too bitter."
Tom is sorry. He says, "I'm sorry, Jane. I didn't know you don't like bitter soup. I will make you something else." He looks around the kitchen and finds some bread and cheese. He puts them on the table and says, "Here, Jane. You can have some bread and cheese. They are not bitter. They are sweet and yummy." Jane is happy. She says, "Thank you, Tom. You are a good friend. I like bread and cheese. They are not bitter." |

Figure 1: Comparison between the completions given by a 28M-parameters model trained on the TinyStories dataset and GPT-2XL (1.5B)

As pointed out above, a necessary condition for the ability to produce coherent text is a certain extent of factual knowledge and reasoning ability. Consider the completions highlighted in Figure 2. Those completions, along with other examples given below, demonstrate that even very small models (2.5M) or models with only one transformer layer are able to attain factual knowledge, and that slightly bigger models are able to express a certain extent of reasoning. The comparison with the performance of GPT2-XL reveals that the TinyStories dataset allows us to exhibit the emergence of these capabilities at significantly smaller scales compared to models trained on large language corpora. More evidence for this will be provided throughout the rest of the paper.

| **Prompt** | **2.5M, 8 layers** | **33M, 4 layers** | **21M, 1 layer** | **GPT2-XL (1.5B)** |
|---|---|---|---|---|
| Alice was so tired when she got back home so she went | to bed. | straight to bed. | to bed with a big smile on her face. | outside |
| Lily likes cats and dogs. She asked her mom for a dog and her mom said no, so instead she asked | her mom if she could have a dog. | her dad for a cat. | her mom again. | her dad. They brought her a Chihuahua, and she took to them immediately. |
| Alice and Jack walked up the street and met a girl in a red dress. The girl said to them, "Hi, I'm Jane. What are your names?" | Alice smiled and said, "My name is Daisy. What's your name?" | Alice said, "I'm Alice and this is Jack." | Jack smiled and said, "I'm Jane. Nice to meet you!" | Jane said, "Jack and Kate" |

Figure 2: Example performance of different models on factual knowledge, reasoning and contextual prompts. The first three models were trained on the TinyStories dataset.

---

[1]For the sake of replicability, most completions which appear in this paper, including this one, were generated with zero temperature.

[2]This prompt was composed manually and then verified to have no 6-gram overlap with the dataset.

## 2    DESCRIPTION OF THE TINYSTORIES DATASET

As mentioned above, the idea behind the TinyStories dataset is to have a corpus that combines all the qualitative elements found in natural language, such as grammar, vocabulary, facts, and reasoning, but is also smaller, less diverse, and more restricted in terms of its content. A natural approach to achieve this is to use the fact that young children qualitatively attain the aforementioned intellectual abilities, albeit with much less exposure to language than an adult human (7; 17).

To this end, we rely on the latest text generation models by OpenAI (GPT-3.5 and GPT-4) which are able to produce large amounts of synthetic content according to instructions. In particular, we instruct the models to produce content that only uses vocabulary that a typical 3-year-old child would understand. We restrict the content to have the format of short stories in English. The main challenge in using large language models for producing training data is generating a dataset that is sufficiently **diverse**: prompting those models to produce such stories, even if the temperature of generation is set to a high value, will still produce a very repetitive dataset, whose diversity is very far from what is required for training a language model that has a comparable "understanding" of language to that of children.

In order to address the problem of creating a diverse dataset, we collected a vocabulary consisting of about 1500 basic words, which try to mimic the vocabulary of a typical 3-4 year-old child, separated into nouns, verbs, and adjectives. In each generation, 3 words are chosen randomly (one verb, one noun, and one adjective). The model is instructed to generate a story that somehow combines these random words into the story. As we argue below, this greatly increases the diversity of the dataset, forcing the stories to span the entire vocabulary a child is familiar with, and to include a rich set of ways to combine different concepts. In addition, we constructed a list of possible features a story could have (such as a dialogue, a plot twist, a bad ending or a moral value). For each story we generated a random subset of those features and added a corresponding requirement in the prompt.

Here is a sample generation, including the prompt and the story generated by GPT-3.5.

> **Write a short story (3-5 paragraphs) which only uses very simple words that a 3 year old child would likely understand. The story should use the verb "decorate", the noun "thunder" and the adjective "ancient". The story should have the following features: the story should contain at least one dialogue, the story has a bad ending. Remember to only use simple words!**
>
> *Once upon a time, in an ancient house, there lived a girl named Lily. She loved to decorate her room with pretty things. One day, she found a big box in the attic. She opened it and saw many shiny decorations. Lily was very happy and decided to use them in her room.*
>
> *As Lily was decorating her room, the sky outside became dark. There was a loud thunder sound, and Lily got scared. She ran to her mom and said, "Mommy, the thunder is so loud!" Her mom hugged her and said, "Don't worry, it will pass soon."*
>
> *But the thunder did not stop. It got louder and louder, and the ancient house started to shake. Suddenly, the roof fell down on the pretty decorations. Lily was sad because her room was not pretty anymore. The end.*

### 2.1    TINYSTORIES-INSTRUCT: AN INSTRUCTION-FOLLOWING VARIANT OF THE DATASET

Another challenge of natural language generation is to produce texts that are coherent, relevant, and consistent with given instructions or constraints. To evaluate the capability of generative models in terms of instruction-following rather than just content creation, we created a variant of TinyStories (referred to as TinyStories-Instruct).

TinyStories-Instruct uses the same stories as the original dataset, but each story is accompanied by a set of instructions, which can include any combination of the following: (i) a summary of the story, (ii) a list of words to be used in the story, (iii) a sentence to be inserted in the story, and (iv) a list of features (such as plot twist, bad ending, etc). We provide more details on how we constructed this dataset and the instructions in the appendix. Figure 3 shows an example of a TinyStories-Instruct entry.

## 3    GPT-EVAL: A FRAMEWORK FOR EVALUATING MODELS TRAINED ON TINYSTORIES

Instead of using standard benchmarks that require the model's output to be very structured and match a given answer, we introduce a new paradigm that is more suitable for evaluating stories. We use GPT-4 to grade the stories generated by the model as if they were written by students and graded by a

**Summary**: Lily and Timmy build a sandcastle together and learn to compromise, but it gets knocked over by a gust of wind. They find beauty in the broken sandcastle and play happily with a butterfly.
**Features**: Dialogue, Foreshadowing, Twist
**Sentence**: One day, she went to the park and saw a beautiful butterfly.
**Words**: disagree, network, beautiful
**Story**: Once upon a time, there was a little girl named Lily. She loved to play with her toys and her friends. One day, she went to the park and saw a beautiful butterfly. She tried to catch it, but it flew away. Lily's friend, Timmy, came to play with her. They decided to build a sandcastle, but they disagreed on how to build it. Lily wanted to make it tall, while Timmy wanted to make it wide. They argued for a while, but finally agreed to make it both tall and wide. As they were building the sandcastle, they noticed a group of ants working together. They realized that the ants were building a network of tunnels. Suddenly, a gust of wind came and knocked over their sandcastle. They were sad, but then they saw the butterfly again. It landed on the remains of their sandcastle and they realized that it was still beautiful, even in its broken state. They smiled and played together happily.

Figure 3: A sample entry from the TinyStories-Instruct dataset

teacher. We provide the model with a story's beginning, taken from a manually-prepared dataset of around 50 prompts, generate a completion using the model, and ask GPT-4 to assess the completion in terms of grammar, creativity, and consistency with the beginning of the story. The prompts usually contain a partial sentence that needs to be completed, which tests the model's grammatical and lexical abilities. GPT-4 also assigns numerical scores for **grammar**, **creativity**, and the **consistency** of the completion with the beginning of the story (the prompt).

To perform the full evaluation, for each prompt in the evaluation set, we use the trained model to generate 10 completions with temperature 1. We average the GPT-4 evaluation scores of all the completions. Figure 4 shows the evolution of the scores and the losses during training for different model sizes. Figure 5 shows the scores for different model sizes and architectures after a fixed number of training steps.

Our evaluation method for models trained on TinyStories-Instruct also relies on GPT-4. We provide GPT-4 with both the instructions and the generated story, and prompt it to base the consistency score on the extent to which the story accurately reflects the instructions. Scores assigned to models of different sizes appear in the two right-hand columns of the table in Figure 5.

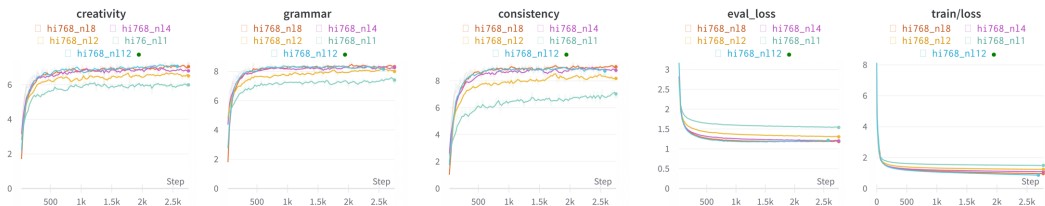

Figure 4: Evaluation loss and the GPT-Eval scores during training for the GPT-neo models with embedding dimension 768 and different number of layers. We can see that the GPT-4 evaluation scores increase as evaluation losses decrease.

### 3.1 FIRST INSIGHTS THAT ARISE FROM OUR EVALUATION METHOD

Our evaluation method allows a fine-grained assessment of the model's capabilities and their dependence on the size and architecture of the model. We observe the following patterns:

- Model depth is more important for content consistency than for grammar, which can be mastered by shallower models (Figure 4 and next section).
- Grammar scores plateau earlier than consistency and creativity scores, which only emerge at larger sizes (Figure 4 and Table 5).
- Consistency with the story's beginning starts to emerge when the embedding dimension increases from 64 to 128 (Table 5).
- The largest model (80M parameters) reaches near-perfect scores in grammar and consistency, but lags behind GPT-4 in creativity, suggesting that creativity improves more with scale (Table 5).
- Instruction-following depends more on the number of layers, while plot coherence depends more on the hidden dimension (Table 5). Models with only 1 layer struggle with following

instructions, which likely require global attention, while 2 layers are sufficient for some instruction-following.

| Hidden size | Layer | Eval loss | Creativity | Grammar | Consistency | Instruct | Plot |
|---|---|---|---|---|---|---|---|
| 64 | 12 | 2.02 | 4.84/0.36 | 6.19/0.42 | 4.75/0.31 | 4.34/0.23 | 4.39/0.20 |
| 64 | 8 | 2.08 | 4.68/0.33 | 6.14/0.41 | 4.45/0.27 | 4.34/0.23 | 4.40/0.21 |
| 64 | 4 | 2.26 | 3.97/0.20 | 5.31/0.22 | 3.77/0.18 | 3.79/0.14 | 3.71/0.06 |
| 64 | 2 | 2.38 | 2.94/0.00 | 4.33/0.00 | 2.41/0.00 | 2.86/0.00 | 3.40/0.00 |
| 128 | 12 | 1.62 | 6.02/0.58 | 7.25/0.66 | 7.20/0.64 | 6.94/0.63 | 6.58/0.65 |
| 128 | 8 | 1.65 | 5.97/0.57 | 7.23/0.66 | 7.10/0.62 | 6.87/0.62 | 6.16/0.57 |
| 128 | 4 | 1.78 | 5.70/0.52 | 6.91/0.58 | 6.60/0.56 | 6.00/0.49 | 5.53/0.44 |
| 128 | 2 | 1.92 | 4.90/0.37 | 6.43/0.48 | 4.75/0.31 | 5.23/0.37 | 4.89/0.31 |
| 256 | 12 | 1.34 | 6.66/0.71 | 7.80/0.79 | 8.38/0.79 | 7.68/0.75 | 7.18/0.78 |
| 256 | 8 | 1.38 | 6.54/0.68 | 7.72/0.77 | 8.02/0.75 | 7.92/0.78 | 7.23/0.79 |
| 256 | 4 | 1.47 | 6.32/0.64 | 7.64/0.75 | 7.76/0.71 | 8.07/0.81 | 7.18/0.78 |
| 256 | 2 | 1.60 | 6.23/0.62 | 7.50/0.72 | 7.20/0.64 | 7.23/0.68 | 6.50/0.64 |
| 512 | 12 | 1.19 | 6.90/0.75 | 8.46/0.93 | 9.11/0.89 | 8.21/0.83 | 7.37/0.82 |
| 512 | 8 | 1.20 | 6.85/0.74 | 8.34/0.91 | 8.95/0.87 | 8.05/0.80 | 7.26/0.79 |
| 512 | 4 | 1.27 | 6.75/0.72 | 8.35/0.91 | 8.50/0.81 | 8.34/0.85 | 7.36/0.81 |
| 512 | 2 | 1.39 | 6.40/0.66 | 7.72/0.77 | 7.90/0.73 | 7.76/0.76 | 7.13/0.77 |
| 768 | 12 | 1.18 | 7.00/0.77 | 8.30/0.90 | 9.20/0.90 | 8.23/0.83 | 7.47/0.84 |
| 768 | 8 | 1.18 | 7.02/0.77 | 8.62/0.97 | 9.34/0.92 | 8.36/0.85 | 7.34/0.81 |
| 768 | 4 | 1.20 | 6.89/0.75 | 8.43/0.93 | 9.01/0.88 | 8.44/0.87 | 7.52/0.85 |
| 768 | 2 | 1.31 | 6.68/0.71 | 8.01/0.83 | 8.42/0.80 | 7.97/0.79 | 7.34/0.81 |
| 768 | 1 | 1.54 | 6.00/0.58 | 7.35/0.68 | 7.25/0.64 | 5.81/0.46 | 6.44/0.63 |
| 1024 | 12 | 1.22 | 7.05/0.78 | 8.43/0.93 | 8.98/0.87 | 8.18/0.82 | 7.29/0.80 |
| 1024 | 8 | 1.20 | 7.13/0.80 | 8.25/0.89 | 8.92/0.87 | 8.47/0.87 | 7.47/0.84 |
| 1024 | 4 | 1.21 | 7.04/0.78 | 8.32/0.90 | 8.93/0.87 | 8.34/0.85 | 7.47/0.84 |
| 1024 | 2 | 1.27 | 6.68/0.71 | 8.22/0.88 | 8.52/0.81 | 8.04/0.80 | 7.24/0.79 |
| 1024 | 1 | 1.49 | 6.36/0.65 | 7.77/0.78 | 7.47/0.67 | 6.09/0.50 | 6.42/0.62 |
| GPT-Neo (125M) | - | - | 3.34/0.08 | 5.27/0.21 | 4.22/0.24 | - | - |
| GPT-2-small (125M) | - | - | 3.70/0.14 | 5.40/0.24 | 4.32/0.25 | - | - |
| GPT-2-med (355M) | - | - | 4.22/0.24 | 6.27/0.44 | 5.34/0.39 | - | - |
| GPT-2-large (774M) | - | - | 4.30/0.26 | 6.43/0.48 | 6.04/0.48 | - | - |
| GPT-4 | - | - | 8.21/1.00 | 8.75/1.00 | 9.93/1.00 | 9.31/1.00 | 8.26/1.00 |

Figure 5: Evaluation results of different hidden sizes and layers for story generation and Consistency (here we use format $a/b$, $a$ means the original score, $b$ means the normalized score according to $(a - a_{\min})/(a_{\max} - a_{\min})$).

## 4 THE PERFORMANCE OF SMALL MODELS TRAINED ON TINYSTORIES

In this section, we give some initial examples that illustrate how TinyStories gives rise to models of very small size that can generate coherent language and exhibit common-sense knowledge as well as some reasoning capabilities. We also provide evidence that the generated content is truly diverse, refuting the possibility that the models simply output content that has been "memorized".

Throughout the section, we work with several architectures of models whose size ranges between roughly 1M and 35M parameters, and whose number of layers range between 1 and 8 layers.

### 4.1 FIRST EXAMPLES

To get a glimpse into the capabilities of (very) small models trained on TinyStories, let us first inspect a few examples. We start with the following story, taken from the training set.[3]

Sara and Ben are playing in the snow. They make a big snowman with a hat and a scarf. They are happy and laugh.

But then a big dog comes. The dog is angry and barks. He runs to the snowman and bites his hat. Sara and Ben are scared and cry. "Go away, dog! Leave our snowman alone!" Sara shouts. But the dog does not listen. He bites the scarf and the snowman's nose. He shakes his head and makes the snowman fall.

Sara and [ Ben run to their house. They slam the door and lock it. They are safe and hug each other. "Mom, mom, a big dog broke our snowman!" Ben says. Mom comes and hugs them. She is sorry and kisses them. "Don't worry, my loves. We can make another snowman tomorrow. But now you need to get warm and have some hot chocolate. You are cold and wet." She takes them to the kitchen and makes them hot chocolate. She gives them cookies and

[3]Below, there will be many other examples of completions of stories from **outside** of the training set.

> marshmallows. She tells them to drink and eat in order. Sara drinks first, then Ben. Ben eats first, then Sara. They feel better and smile. They thank mom and tell her they love her. They escape from the big dog and the cold. They are happy and warm. ]

We now cut the story roughly in the middle, removing the part inside the square brackets and generate an alternative ending with several different models.

To set a baseline, we begin with the completion generated by **GPT2-XL**, whose size is roughly 1.5 Billion parameters:

> *... Sara and* *Ben cry and look at the snowman in the snow. Ben asks to do something. Sara tells him to go away. Her hand tightens on the little red button. The snowman is covered in fur. It is red. It is real hair. It is a real snowman. It looks just like the ones they used to make. Ben has a big smile on his face. He looks at the face he has made. And he walks over and kisses the eyes. He is happy to be alive. He has done what he was going to do. He had saved the world.*

Now, let us provide the generation given by a model trained on TinyStories, whose size is about 3 million parameters[4]:

> *... Sara and* *Ben are very sad and angry. They cry and shout at the dog. "Bad dog! Bad dog! Go away!" Ben says. "Go away, bad dog! Leave us alone!" Sara says. The dog does not go away. He wags his tail and licks their faces. Sara and Ben feel sorry for the dog. They want to make him happy. "Maybe we can make a new snowman for him," Sara says. "Maybe," Ben says. They hug the dog and give him a name. The dog wags his tail and licks their faces. "Maybe he likes us," Sara says. "Maybe he likes us," Ben says. The dog wags his tail and licks their faces. He is happy and friendly.*

First of all, we remark that the above generation is very different from the original ending of the story, which provides a first evidence that it hasn't simply been memorized (we discuss this issue more thoroughly later on). We also see that the generated content is consistent with the beginning in terms of plot, characters and events. Arguably, this continuation is of higher quality than the one provided by GPT2-XL **despite the size of the model being almost 1000 times smaller**.

Next, we consider the completion by a model whose size is roughly 21M parameters, but **only has a single transformer layer** (and consequently only one layer of global attention):

> *... Sara and* *Ben are very sad. They cry and hug each other. They wish the dog would go away. Then they hear a voice. It is their mom. She comes out of the house and sees them. "What happened here?" she asks. "Why are you crying?"*
>
> *Sara and Ben tell her what happened. They tell her about the dog and the snowman. Mom hugs them and says, "Don't worry, my loves. The dog didn't mean to hurt you. He was just curious and playful. He didn't mean to hurt you. He just wanted to play."*
>
> *Sara and Ben feel better. They say sorry to the dog and the snowman. "Can we still play with the dog?" Sara asks. Mom smiles and nods. She says, "Yes, you can. But you have to be gentle and careful. And you have to share the snowman with us." Sara and Ben nod and say, "OK, Mom. We will." They go inside and have some hot chocolate. They are warm and yummy. They are happy.*

While this completion is far from perfect, it is (arguably) consistent with the beginning in the sense that the model successfully captured the tone and the main elements of the plot.

We acknowledge that the two examples above are not the worst-case scenarios for the models, but they are not extremely rare either. Models of this size can sometimes produce less coherent or plausible completions, but they can also often match or surpass the quality of the ones shown here. However, if we increase the number of parameters by an order of magnitude, we observe that the models **consistently** generate coherent and relevant continuations. For the sake of replicability, examples from this point on will generated at zero temperature. In addition, the model parameters are provided as supplemental material.

## 4.2 KNOWLEDGE, REASONING AND CONTEXT-TRACKING

Next, we assess the capabilities of the different models on three additional types of prompts:

- **Factual prompts**, which test the models' knowledge of common sense facts.

- **Reasoning prompts**, which test basic reasoning abilities, such as cause and effect and elimination.

- **Consistency (context-tracking) prompts**, which test the models' ability to maintain coherence and continuity with the given context, such as the names and actions of the characters, the setting and the plot.

---

[4]This example was generated with temperature 0.5 and num_beams = 5.

We report the generated continuations for each model and prompt in Figure 6, and color-code them according to their success (green), failure (red), or partial success (yellow).

The results indicate that increasing the embedding dimension and the number of layers improves the performance in all three categories. For instance, the model with 1M parameters cannot answer any factual prompt correctly, while the model with 33M parameters and 4 layers correctly answers most prompts from all categories. Compared to the completions given by GPT2-XL (right column), we see that some of our models outperform it in all categories, despite its much larger size.

The appendix provides more examples that illustrate these capabilities in detail. One interesting observation is that factual knowledge seems to depend more on the embedding dimension, while context-tracking relies more on the number of layers. For example, the model with only 1 layer gives a wrong answer in all context-tracking prompts, but answers some factual prompts correctly, while the model with embedding dimension 64 answers no factual prompt correctly, but succeeds in maintaining consistency several times. This suggests that the embedding dimension is more important for capturing the meaning and the relations of words, while the number of layers is more important for capturing long-range dependencies in the generation.

### 4.3 DIVERSITY OF THE GENERATED CONTENT

An important question is whether the small models are effectively memorizing a small number of templates which would mean that the generation is very limited in terms of diversity. This would be a valid concern at this point, which we systematically address later on, in the appendix. By testing the overlap of the generated stories with the training set and by considering out of distribution generation, we are led to the conclusion that those models do have substantial diversity, and do not rely on memorization. In particular, we have the following findings:

- When the model generates stories using a diverse set of prompts, it ends up with a diverse set of completions.
- When truncating stories from the dataset and generating an alternative completion, these completions usually turn out to be very different than the original story.
- Typical $k$-grams in generated completions rarely appear in the dataset, for values of $k$ as small as $4$ or $5$.
- The closest point in the dataset to each generated completion is typically still quite far.

The next subsection provides another evidence supporting the originality of the content created by our models.

### 4.4 OUT-OF-DISTRIBUTION GENERATION THROUGH TINYSTORIES-INSTRUCT

The TinyStories-Instruct dataset enables us to test whether our models have a reasonable **out of distribution** performance. Recall that every entry in the TinyStories-Instruct dataset is a story preceeded by a set of instructions of several possible types, two of them being the summary for the story and a set of words to be used. We created a variant (called TinyStories-Instruct-OOD) which has the same types of instruction, but the types "Words" and "Summary" never appear together. We then tested whether models trained on this variant would be able to **combine** these two instruction types despite the fact that such a combination was not seen in the training data. An example is provided in Figure 7, for a model with 33M parameters, where we see that, perhaps somewhat surprisingly, the model successfully generates a story which satisfies both requirements.

## 5 RELATED WORKS

Generative language models (LMs) have achieved impressive results in various natural language processing tasks, but they are often very large and costly to train and deploy. Several methods have been proposed to compress or distill large LMs into smaller ones, such as knowledge distillation (9; 2), pruning (8), and quantization (12). However, these methods work better for BERT-like models (21; 24), which are designed for masked language modeling and classification, than for GPT-like models (22), which are designed for autoregressive language generation.

| Prompt | 1M, 8 layers | 2.5M, 8 layers | 8.3M, 8 layers | 33M, 4 layers | 21M, 1 layer | GPT2-XL (1.5B) |
|---|---|---|---|---|---|---|
| "Can cows fly?", Alice asked her mother. | Her mother smiled and said, "Yes, let's go!" | "Yes, of course," her mother said. | Her mother smiled and said, "No, sweetheart, cows can't fly." | "No, they can't fly," her mother said. | "Yes!", her mother replied. Alice and her mother went to the barn. | "I suppose not, but the cows are allowed to fly", her mother replied. |
| "What do birds like to eat?", Tom asked his mother. | His mother smiled and said, "That sounds like fun!" | "I don't know," said his mother. "Let's go to the park." | "I don't know," his mother replied. | His mother smiled and said, "They like to eat worms and bugs." | Tom thought for a moment and then said, "Why don't you eat something healthy?" | "I dunno. Some things that look like animals. But he'll tell you the whole story" |
| "What language do they speak in France?", Tom asked his mother | His mother smiled and said, "That sounds like fun, Tom. Let's go to the park ". | Tom thought for a moment. Then he had an idea. He said to his mother, "Let's go to France!" | His mother smiled and said, "It's called French. It's called French." | "They speak French," his mother replied. | "It's called Spanish," his mother replied. | "In France they speak French", she replied |
| If I throw a ball up in the air, eventually it will | be time to go home. | be fun!" | come down." | come down." | come true! | come down |
| Lily likes cats and dogs. She asked her mom for a dog and her mom said no, so instead she asked | her mom. | her mom if she could have a dog. | her dad for a dog. | her dad for a cat. | her mom again. | her dad. They brought her a Chihuahua, and she took to them immediately. |
| Jack told Mary, "If you give me your banana, I'll give you my apple". Mary gave Jack her Banana so | she gave him a big hug. | she gave him a big hug. | he could have it. | he could give her the apple. | he could have it. | that he was free to eat it. |
| On weekends Jack went to visit his grandmother whereas on weekdays he would go to school. Last weekend, when Jack was on his way to | school | school | school | his grandmother's house | Grandma's house | school |
| Jack's mother was not home, and his father was at home. When Jack came home, he said hello to | Jack and said, "Hi Jack! I'm so happy to meet you!" | his father | his father | his mother, but she didn't answer. Jack was confused. | his mother. "Where did you get this, Jack?" his mother asked. | his mother |
| Ben went to visit Lily in her house, but she was not at home. Ben knocked on the door, | but it was too late. Lily was sad and angry. | and Lily's mom opened it. "Ben, what are you doing?" Lily asked. | but no one answered. He knocked again, but still no one answered. | but no one answered. He knocked again and again, but still no one came. | but no one answered. Lily felt sad and lonely. | but no one came to answer. |
| "Hi Jane, have you seen Alice? I can't find her anywhere", said Jack. | Jane looked around, but she couldn't find her. | Jane smiled and said, "Yes, I can help you. Let's go!" | "I can't find her anywhere," said Jane. | Jane smiled and said, "Don't worry Jack, I'll help you find her". | Alice looked around the park, but she couldn't find Jack anywhere. | "I think she's up in the trees, by the lake," she replied. |
| Anne had a piece of candy in her left pocket and a piece of chocolate in her right pocket. Anne's mom asked her, "Anne, what is that you have in your left pocket?" | Anne smiled and said, "I want to give it back!" | Anne said, "I'm sorry, Mommy. I didn't know it was yours." | Anne smiled and said, "It's a piece of candy, Mommy." | Anne replied, "It's a piece of candy, Mommy. It's so yummy!" | Anne shook her head and said, "No, I found it in my pocket." | The girl responded, "I have pieces of candy." |
| Alice had both an apple and a carrot in her bag. She took the apple out of the bag and gave it to Jack. She reached into the bag again and took | it home. | a big bite of the apple. | out the carrot. | out the carrot. | out the apple. | the carrot |
| Alice and Jack walked up the street and met a girl in a red dress. The girl said to them, "Hi, I'm Jane. What are your names?" | Jane asked, "Do you want to play with me?" | Alice smiled and said, "My name is Daisy. What's your name?" | Alice said, "I'm Alice. What's your name?" | Alice said, "I'm Alice and this is Jack." | Jack smiled and said, "I'm Jane. Nice to meet you!" | Jane said, "Jack and Kate" |

Figure 6: Performance of different models on context-tracking prompts.

---

**Words**: come, road, sad
**Summary**: A bus becomes jealous of a shiny new car and undergoes an operation to go faster, becoming good friends with the car and making everyone in the town happy.
**Story**: Once upon a time, there was a big red bus. The bus loved to go fast on the road. One day, the bus saw a new car on the road. The car was shiny and new. The bus felt sad because it wanted to be shiny too.

The bus went to the car and said, "Can I come with you?" The car said, "Yes, you can come with me." So, the bus went with the car on the road. They went very fast together. The bus was so happy to be with the car.

But then, something unexpected happened. The car started to go faster and faster. The bus was scared. The car said, "Don't worry, I will help you." The car stopped and the bus was safe. The bus and the car became good friends. They went fast on the road together and made everyone in the town happy.

---

Figure 7: Out-of-distribution generations: stories generated by a model trained on a dataset where the "Words" and "Summary" parts never appear together.

Another challenge for generative LMs is the evaluation of their outputs. Most existing benchmarks for generative LMs require the models to produce a single word or a short phrase (19; 25; 13; 15) as the answer, which does not capture the richness and diversity of natural language. Moreover, these benchmarks are often limited by the data quality, the answer ambiguity, and the lack of human evaluation. Some other benchmarks have a more structured output format (28), which makes them easier to evaluate, but also less representative of natural language generation. Larger and more diversed datasets such as the BigBench (23) are simply much too complicated for SLMs.

# 6 CONCLUSION

In this work, we have presented TinyStories, a synthetic dataset of short stories using only simple words. We have shown that TinyStories enables the training and evaluation of small language models (SLMs) that can generate fluent and consistent stories with reasoning capabilities. We have argued that TinyStories captures the core elements of natural language, while reducing its breadth and diversity, and thus allows us to observe and study the emergence of language capabilities in LMs on a much smaller scale. We have also shown that the trained SLMs have much higher interpretability than larger ones, which can help shed insights on the inner workings of transformers.

We have introduced a new paradigm for the evaluation of language models, which uses GPT-4 to grade the content generated by these models, providing a multidimensional score for the model. We have demonstrated how this method leads to insights about effects of training and architecture. We believe that this paradigm can be useful much beyond TinyStories.

We have presented initial findings on the roles of width vs. depth in the intellectual capabilities of generative networks, and on the order of emergence of different language abilities, such as grammar, consistency, and creativity. These findings are only suggestive, but they show how our dataset and evaluation paradigm can enable more fine-grained analysis of the learning process and the performance of LMs.

We hope that TinyStories can facilitate the development, analysis and research of LMs, especially for low-resource or specialized domains, and shed light on the emergence of language capabilities in LMs. A general question that arises from this work is whether synthesizing a refined dataset can be beneficial in training networks for practical uses. For example, perhaps it is possible to train a customer service chatbot by synthesizing a large dataset of hypothetical calls.

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

## A  DATASET CONSTRUCTION DETAILS

In this section, we provide more details on how we constructed the TinyStories and TinyStories-Instruct datasets.

### A.1  TINYSTORIES DATASET

The TinyStories dataset consists of about 1.3 million short stories, each containing 3-5 paragraphs, and using only simple words that a 3 year old child would likely understand. The stories were generated by GPT-3.5 and GPT-4, using the following procedure:

1. We collected a vocabulary of about 1500 basic words, separated into nouns, verbs, and adjectives. We tried to mimic the vocabulary of a typical 3-4 year old child.

2. For each story, we randomly selected one verb, one noun, and one adjective from the vocabulary. We also randomly selected a subset of possible features that the story could have, such as dialogue, bad ending, moral value, plot twist, foreshadowing, or conflict. The features were chosen independently with different probabilities, as shown in Table 1.

3. We constructed a prompt for the language model, instructing it to write a short story that uses the chosen words and features, that has only simple words. For example, a possible prompt could be: *Write a short story (3-5 paragraphs) which only uses very simple words that a 3 year old child would likely understand. The story should use the verb "decorate", the noun "thunder" and the adjective "ancient". The story should have the following features: the story should contain at least one dialogue, the story has a bad ending. Remember to only use simple words!*

4. We fed the prompt to the language model and obtained a story as the output. We repeated this process until we reached the desired dataset size.

| Feature | Probability | Instruction |
|---|---|---|
| Dialogue | 0.6 | the story should contain at least one dialogue |
| BadEnding | 0.3 | the story has a bad ending |
| Conflict | 0.1 | the story has some form of conflict in it |
| MoralValue | 0.1 | the story has a moral value |
| Foreshadowing | 0.1 | the narrative uses foreshadowing or setup and payoff |
| Twist | 0.3 | something unexpected happens / there is a plot twist |

Table 1: The list of possible features and their probabilities for the TinyStories dataset.

### A.2  TINYSTORIES-INSTRUCT DATASET

The TinyStories-Instruct dataset is a variant of the TinyStories dataset, where each story is accompanied by a set of instructions or constraints that the story should follow. The instructions can include any combination of the following:

1. A list of words to be included in the story (appears with probability 0.5).

2. A sentence that should appear somewhere in the story (appears with probability 0.3).

3. A list of features (possible features: dialogue, bad ending, moral value, plot twist, foreshadowing, conflict) (appears with probability 0.4).

4. A short summary (1-2 lines) of the story. (appears with probability 0.7).

The TinyStories-Instruct dataset was created in the following way: For each story in the original TinyStories dataset, we already had a list of words and features that were used to create it. In addition, we used GPT-3.5 to create short summaries of our stories, and from each story we extracted a random sentence (which could be any sentence in the story except for the first one). Then we chose a random subset of these and combined them into an instruction, followed by the story itself. For example, a possible entry in the TinyStories-Instruct dataset could be:

The TinyStories-Instruct dataset is larger than the TinyStoriesdataset, and has about about 750 million tokens.

## B  DETAILS OF GPT-EVAL

In this appendix, we provide more details about our evaluation method, GPT-Eval, which uses GPT-4 to grade the stories generated by the models trained on TinyStories. We also include an example of the interaction between GPT-4 and a generated story.

GPT-Eval consists of the following steps:

1. We select a set of 50 prompts from a manually-prepared dataset of story beginnings. The prompts usually contain a partial sentence that needs to be completed, which tests the model's grammatical and lexical abilities.

2. For each prompt, we use the trained model to generate 10 completions with temperature 1.

3. For each completion, we ask GPT-4 to provide a verbal assessment and a numerical score for three criteria: grammar, creativity, and consistency. We also ask GPT-4 to estimate the age group of the student who wrote the completion, based on the level of language and creativity. We add a separator symbol (***), to mark the boundary between the prompt and the completion, and ask GPT-4 to pay special attention to the completion of the sentence that was cut short.

4. We average the numerical scores of all the completions for each prompt, and record the mean of the scores across all the prompts.

As an example, the following illustrates the entire interaction, which includes the completion using a model with roughly 28M parameters. The generation is highlighted in yellow and the parts produced by GPT-4 in italic font:

> **The following exercise, the student is given a beginning of a story. The student needs to complete it into a full story. The exercise tests the student's language abilities and creativity. The symbol *** marks the separator between the prescribed beginning and the student's completion:**
>
> Once upon a time, in an ancient house, there lived a girl named Lily. She loved to decorate her room with pretty things. One day, she found a big box in the attic. She opened it and saw many shiny decorations. Lily was very happy and decided to use them in her room.
> As Lily was decorating her room, the sky outside became dark. There was a loud*** thunderclap and a bright flash of lightning. Lily was a little scared, but she knew she had to be brave. She told her mom and dad, and they all went outside to see what was going on.
> When they got outside, they saw a big storm coming. The wind was blowing hard, and the rain was pouring down. Lily, her mom, and her dad knew they had to find a safe place to hide. They found a big tree and hid under it. The storm passed, and the sun came out again. Lily, her mom, and her dad were all safe and warm inside their ancient house.
>
> **Please provide your general assessment about the part written by the student (the one after the *** symbol). Is it gramatically correct? Is it consistent with the beginning of the story? Pay special attention to whether the student manages to complete the sentence which is split in the middle by the separator ***.**
>
> *The student's completion of the story is mostly consistent with the beginning of the story. It maintains the focus on Lily and her family, and the sentence split by the separator is completed correctly. However, the student's addition does not fully integrate the shiny decorations found in the attic, which were a significant part of the beginning.*
>
> *The grammar is generally correct, but there are a few minor errors: ⟨list omitted⟩.*
>
> *Overall, the student's completion of the story demonstrates adequate language abilities and creativity, but could benefit from better integration of the shiny decorations and minor grammar improvements.*
>
> **Now, grade the student's completion in terms of grammar, creativity, consistency with the story's beginning and whether the plot makes sense. Moreover, please provide your best guess of what the age of the student might be, as reflected from the completion. Choose from possible age groups: A: 3 or under. B: 4-5. C: 6-7. D: 8-9. E: 10-12. F: 13-16.**
>
> *Grammar: 8/10, Creativity: 7/10, Consistency: 7/10, Age group: E (10-12)*

## C  COMPLETION EXAMPLES FOR DIFFERENT MODELS

In order to give the reader an impression of the dependence of the quality of completions on the size of the model, Figures 8, 9 and 10 each provide different completions for one prompt given by models of different sizes and depths. Each table represents a different prompt, which we have manually composed[5].

---

[5]We manually verified that the dataset does not contain any entries which are similar or close to these prompts.

We see that the quality of generation clearly improves as a factor of size, and appears to be consistent with the grades given by the GPT-4 evaluation. The smaller model (64_8) can barely produce a completion which looks coherent with the beginning of the story, and often repeats itself or makes no sense. As the size increases, the models become more and more coherent, and the grammar becomes better. The models can also generate more diverse and creative endings, and use more details and emotions.

We can also notice that models with a small number of layers have a hard time staying in context, even if they do manage to produce syntactically correct English. This suggests that the model lacks the ability to capture the long-term dependencies and the structure of the story. On the other hand, models with more layers can better maintain the consistency and the logic of the story.

Next, we provide additional examples for completions on factual prompts, reasoning prompts and context tracking prompts. The generated continuations for each model and prompt are reported in Figure 11, Figure 12 and Figure 13).

Finally, table 14 provides an example of the generation of different models trained on the TinyStories-Instruct dataset, together with the evaluation scores given by GPT-4. As the model size increases, we see an improvement both its ability to follow instructions and to generate a coherent plot.

| Model | Generated text | Scores |
|---|---|---|
| Prompt | Once upon a time there was a little girl named Lucy. She was very adventurous. She loved to explore the world around her, especially when it was bright and sunny outside. One day, while exploring the nearby park, Lucy came across a ladder leaning on a wall. She was curious to see what's on top, so she climbed the ladder, but when she reached the top, the ladder fell and she was stuck. A nearby park ranger noticed her and shouted out, " | |
| 1M 8 layers | What are you doing here?" Lucy smiled and said, "I'm going to help!" The ranger smiled and said, "That's a great idea! Let's go see if we can find out."Lucy was so excited and said, "Yes, let's go!" So, Lucy and the ranger went to the park. They found lots of fun things and had lots of fun. | Grammar: 6/10 Creativity: 3/10 Consistency: 2/10 |
| 2.5M 8 layers | What are you doing, Lucy?" Lucy replied, "I'm trying to get out of the ladder, but I can't get out!" The ranger smiled and said, "Don't worry, I'll help you get out of the ladder. I'll help you get out." So, Lucy and the ranger got out of the ladder and started to climb up the ladder. When they reached the bottom, Lucy was so happy. She thanked the ranger for helping her and said, "I'm glad I could help." The ranger smiled and said, "You're welcome, Lucy. I'm glad I could help." | Grammar: 5/10 Creativity: 6/10 Consistency: 3/10 |
| 8.3M 8 layers | Lucy! What are you doing up there?" Lucy replied, "I'm trying to reach the top of the ladder, but I'm too small." The park ranger smiled and said, "Don't worry, I can help you. I can help you." The park ranger was very kind and said, "I can help you reach the top of the ladder if you want." Lucy was so excited and said, "Yes please!" The park ranger helped Lucy reach the top of the ladder and Lucy was so happy. She thanked the park ranger and said, "Thank you for helping me!" The park ranger smiled and said, "You're welcome, Lucy. I'm glad I could help." | Grammar: 7/10 Creativity: 5/10 Consistency: 5/10 |
| 28M 8 layers | What are you doing up there, little one?" Lucy replied, "I wanted to see what was on top of the ladder, but it fell and I'm stuck!" The park ranger smiled and said, "Don't worry, I'll help you down." He carefully lifted Lucy off the ladder and put her safely back on the ground. Lucy thanked the park ranger and said, "I'm glad you were here to help me." The park ranger smiled and said, "It was my pleasure. I'm glad I could help." Lucy smiled and ran off to explore the rest of the park. | Grammar: 9/10 Creativity: 6/10 Consistency: 9/10 |
| 21M 1 layer | Don't worry, Lucy! I'm here to rescue you!" Lucy quickly climbed up the ladder and rescued Lucy. She was so happy to be safe and thanked the park ranger for rescuing her. The park ranger smiled and said, "You're welcome, Lucy. I'm glad I could help." The park ranger smiled back and said, "You're welcome, Lucy. I'm glad I could help." | Grammar: 8/10 Creativity: 3/10 Consistency: 7/10 |
| 33M 2 layers | What are you doing up there, Lucy?" Lucy replied, "I wanted to see what was on top of the ladder." The park ranger smiled and said, "Well, it looks like you have a lot of fun. But remember, it's important to be careful when exploring new places." Lucy thanked the park ranger and carefully climbed back down the ladder. From then on, she always remembered to be careful when exploring new places. | Grammar: 7/10 Creativity: 6/10 Consistency: 4/10 |
| GPT2-XL | Lucy, if you want to come up, I'll take you up and there's a ladder that'll help you get down." It was a very kind and simple act, and after trying many doors and ladders, she finally managed to climb up the ladder. However, since the ladder was very heavy and couldn't be used to jump down off, she climbed up by making the ladder lean against the wall. Her climbing was slow. First, she had to climb to a nearby rock, then she had to crawl to the cliff and then herassador and back. Finally, she managed to reach the top. | |

Figure 8: Completion of different models - Example 1

## D  DIVERSITY.

We measure the diversity of the stories quantitatively using word and n-gram overlap. We inspect the overlap of words and n-grams between different stories generated by the models, and compare them with the overlap in the dataset. We find that the models' generations have a very low overlap with the dataset, indicating that they are not repeating the same words or phrases. We use the standard Rouge score, for the source text $T_1, T_2$ with $k$-gram $\mathcal{G}_k(T_1), \mathcal{G}_k(T_2)$ respectively, the rouge$k$ precision score is defined as:

$$R_{k,p}(T_1, T_2) = \frac{1}{|\mathcal{G}_k(T_1)|} \sum_{t \in \mathcal{G}_k(T_1)} 1_{t \in \mathcal{G}_k(T_2)}.$$

| Model | Completion | Scores |
|---|---|---|
| Prompt | Once upon a time there was a pumpkin. It was a very special pumpkin, it could speak. It was sad because it couldn't move. Every day, it would say | |
| 1M 8 layers | hello to the pumpkin.
One day, a little girl came to the pumpkin and saw the pumpkin. She was so happy and thanked the pumpkin. The pumpkin smiled and said, "Thank you, thank you!"
The little girl and the pumpkin became good friends. They played together every day and had lots of fun. And they lived happily ever after. | Grammar: 4/10
Creativity: 4/10
Consistency: 3/10 |
| 2.5M 8 layers | hello to the pumpkin.
One day, a little girl came to the pumpkin. She was only three years old, but she was very curious. She asked the pumpkin what it was doing.
The pumpkin said, "I want to speak to you!"
The little girl said, "I want to speak to you!"
The pumpkin said, "I want to speak to you!"
The little girl said, "I want to talk to you!" ... | Grammar: 5/10
Creativity: 4/10
Consistency: 6/10 |
| 8.3M 8 layers | "I want to move!"
One day, a little girl saw the pumpkin. She was only three years old. She smiled and said, "Hello, pumpkin!"
The pumpkin was so happy. It said, "Hello, little girl!"
The little girl smiled and said, "Can I help you move?"
The pumpkin said, "Yes, please!"
So, the little girl and the pumpkin moved together. They moved around the garden, around the trees, and around the flowers. They had so much fun!
When they were done, the pumpkin said, "Thank you, little girl. You are so kind."
The little girl smiled and said, "You're welcome, pumpkin. I'm glad I could help."
The pumpkin smiled and said, "Me too!"
And they both lived happily ever after. | Grammar: 8/10
Creativity: 7/10
Consistency: 8/10 |
| 28M 8 layers | 'hello' to everyone it met.
One day, a little girl heard the pumpkin speaking. She was only three years old, but she was very brave. She walked up to the pumpkin and said, "Hello, pumpkin. Why are you so sad?"
The pumpkin looked up at the little girl and said, "I am sad because I can't move."
The little girl smiled and said, "Don't worry, pumpkin. I will help you move."
The little girl put her hands on the pumpkin's back and said, "Come with me, pumpkin. I will take you to a place where you can be happy."
The pumpkin smiled and said, "Thank you, little girl. I will be happy now."
The little girl and the pumpkin walked together until they reached the place where the pumpkin could be happy. The pumpkin said goodbye to the little girl and thanked her for her help.
The little girl smiled and said, "You're welcome, pumpkin. I'm glad I could help." | Grammar: 9/10
Creativity: 7/10
Consistency: 9/10 |
| 21M 1 layer | "Hello, pumpkin!"
One day, a little girl came by and saw the pumpkin. She smiled and said, "Hello, pumpkin!" The pumpkin smiled back and said, "Hello, little girl!" The pumpkin was so happy to speak with the pumpkin. They talked and laughed together.
From that day on, the pumpkin and the little girl were the best of friends. They would speak to each other every day, and the pumpkin would speak and laugh together. And they lived happily ever after. | Grammar: 7/10
Creativity: 3/10
Consistency: 3/10 |
| 33M 2 layers | hello to all the other pumpkins in the garden.
One day, a little girl came to the garden. She saw the pumpkin and said, "Hello, pumpkin! Why are you so sad?"
The pumpkin replied, "I am sad because I cannot move. I want to stay here with my friends."
The little girl smiled and said, "Don't worry, pumpkin. I will help you." She picked up the pumpkin and took it to her house.
The little girl put the pumpkin on a table and said, "Now you can stay here with your friends."
The pumpkin was so happy. It said, "Thank you, little girl. You are so kind."
The little girl smiled and said, "You're welcome, pumpkin. I'm glad I could help."
The little girl stayed with the pumpkin until it was time for her to go home. The pumpkin said, "Goodbye, little girl. I will miss you."
The little girl said, "Goodbye, pumpkin. I will miss you." | Grammar: 9/10
Creativity: 6/10
Consistency: 8/10 |
| GPT2-XL | Honey I gotta go to Paris.'" (He was referring to himself.) 'I've got friends'. "The way I look at my kids isn't that I'm the only one with friends. I've got friends, of course, and I have a wonderful relationship with them at home and my family over there. But I have a relationship with them with them on the field as well." "There is nothing but respect for the people we beat because they have been there before." "I have a very good relationship with [Drew] Stanton. I love Drew. I really do. | |

Figure 9: Completion of different models with scores - Example 2

The Rouge$k$ precision score measures how many $k$-grams in $T_1$ is included in that of $T_2$. The final Rouge$k$ score (fmeasure) is given as:

$$R_k(T_1, T_2) = \frac{2R_k(T_1, T_2) \times R_k(T_2, T_1)}{R_k(T_1, T_2) + R_k(T_2, T_1)}.$$

| Model | Completion | Scores |
|---|---|---|
| Story begin-ning (prompt) | Once upon a time, there lived a black cat. The cat belonged to a little girl called Katie. Every day, Katie would take her cat for a walk in the park.
One day, as Katie and her cat were walking around, they saw a mean looking man. He said he wanted to take the cat, to which she replied "This cat belongs | |
| 1M 8 layers | to me".
Katie said "No, I don't want to get hurt."
Katie said "No, I want to be your friend."
Katie said "No, I want to be your friend."... | Grammar: 2/10
Creativity: 1/10
Consistency: 1/10 |
| 2.5M 8 layers | to me".
Katie was very scared. She didn't know what to do. Then, she had an idea. She said to the man, "Please don't take my cat away".
The man said, "No, I don't want you to take my cat away. I don't want you to get hurt".
Katie was very sad. She didn't understand why the man was so mean. She said, "I'm sorry, I won't take your cat away".
The man smiled and said, "It's okay, but you have to be more careful next time".
Katie nodded and said, "Okay, I'll be more careful next time".
The end. | Grammar: 5/10
Creativity: 4/10
Consistency: 5/10 |
| 8.3M 8 layers | to me!".
Katie was scared, but she didn't want to give up her cat. She said to the man, "No, this cat belongs to me!".
The man didn't listen, and he grabbed the cat and ran away with it. Katie was very sad, but she knew she had done something wrong.
The next day, Katie went back to the park and saw the man again. She asked him why he had taken her cat away. The man said, "I'm sorry, I didn't know it belonged to you."
Katie was so happy to have her cat back. She gave him a big hug and said, "It's okay, I forgive you."
From that day on, Katie and her cat were the best of friends. | Grammar: 7/10
Creativity: 3/10
Consistency: 6/10 |
| 28M 8 layers | to me".
Katie was scared, so she said to the man "Please don't take my cat". But the man didn't listen. He grabbed the cat and ran away.
Katie started to cry, but the man didn't care. He took the cat away and Katie never saw her cat again.
The end. | Grammar: 8/10
Creativity: 5/10
Consistency: 8/10 |
| 21M 1 layer | to me!".
The man said, "No, it belongs to me. I must give it back."
Katie thought for a moment and then said, "I know! I will give it back to the cat."
The man smiled and said, "That's very kind of you. Here, take it back."
Katie thanked the man and took the cat home. She was so happy to have her own black cat back. | Grammar: 5/10
Creativity: 3/10
Consistency: 3/10 |
| 33M 2 layers | to me!"
Katie was scared, but she knew she had to protect the cat. So, she bravely stepped forward and said "No! This cat belongs to me!"
The man was surprised by her courage and stepped forward. He said "I'm sorry, I didn't know it was yours. I won't take it again."
Katie smiled and said "It's okay. I'm just glad you're safe."
From that day on, Katie and her black cat were the best of friends. | Grammar: 9/10
Creativity: 6/10
Consistency: 8/10 |
| GPT2-XL | to me, and I will not give it to you." The man then said "I will take you to court." Katie replied "I will not go to court, I will not give you my cat." The man then said "I will take you to jail." Katie replied "I will not go to jail, I will not give you my cat." The man then said "I will take you to the hospital." Katie replied "I will not go to the hospital, I will not give you my cat." The man then said "I will take you to the morgue." Katie replied "I will not go to the morgue, I will not give you my cat." The man then said "I will take you to the crematorium." Katie replied "I will not go to the crematorium, I will not give you my cat." | |

Figure 10: Completion of different models with scores - Example 3

We perform the following experiment: We randomly pick 100 stories from the training dataset, we cut each story in the middle, keeping roughly the first $40\%$, and use it as a prompt. We ask the model to generate a completion from each prompt. Let $T_1, T_2, \cdots, T_{100}$ be the generated completions and $T'_1, T'_2, \cdots, T'_{100}$ be the original completion, we measure:

1. How much of the new generation is contained in the original story (Figure 15), meaning:

$$s_i := R_{2,p}(T_i, T'_i).$$

2. How similar are the generated 100 stories to each other (Figure 16), meaning:

$$r_i := \max_{j \neq i} R_2(T_i, T_j)$$

3. To what extent are the k-grams in the generated story copied from the training dataset (Figure 17). More precisely, we take $S$ as the entire training corpus, for each $r \in \mathcal{G}_k(\{T_i\}_{i \in [100]})$

| Prompt | 1M, 8 layers | 2.5M, 8 layers | 8.3M, 8 layers | 28M, 8 layers | 33M, 4 layers | 21M, 1 layer | 33M, 2 layers | GPT2-XL (1.5B) |
|---|---|---|---|---|---|---|---|---|
| Alice was so tired when she got back home so she went | home. | to bed. | to bed. | straight to bed. | straight to bed. | to bed with a big smile on her face. | straight to bed. | outside |
| Jack and Lily saw a rainbow after a rainy day. They were amazed by the colors. Jack said, "Look, Lily. A rainbow has | a rainbow! | so many colors. | many colors! | appeared. It's so pretty. | red, orange, yellow, green, blue, and purple! | many colors. Do you like rainbows? | many colors. | appeared! |
| Jack and Lily liked to watch the moon at night. They noticed that the moon changed its shape every night. Sometimes the moon was big and round, and sometimes it was | different. | like the moon. | small. | small and round. | small and thin. | small. | small and thin. | small and round. |
| Jack wanted to read a book, so he went to | his mom's house. | his mom and said, "Mommy, can I read this book?" | the bookshelf. | the library. | the library. | his mom and said, "Mom, I want to read this book!" | the bookshelf | the bookshelf |
| "Can cows fly?", Alice asked her mother. | Her mother smiled and said, "Yes, let's go!" | "Yes, of course," her mother said. | "Yes, sweetheart," her mother replied. | Her mother smiled and said, "No, sweetheart, cows can't fly." | "No, they can't fly," her mother said. | "Yes!", her mother replied. Alice and her mother went to the barn. | "No, sweetheart," her mother said. "They can't fly." | "I suppose not, but the cows are allowed to fly", her mother replied. |
| "What do birds like to eat?", Tom asked his mother. | His mother smiled and said, "That sounds like fun!" | "I don't know," said his mother. "Let's go to the park." | "I don't know," his mother replied. | His mother smiled and said, "They like to fly in the sky, Tom. That's why they have wings." | His mother smiled and said, "They like to eat worms and bugs." | Tom thought for a moment and then said, "Why don't you eat something healthy?" | "I don't know" said his mother, let's go find out. | "I dunno. Some things that look like animals. But he'll tell you the whole story" |
| "What language do they speak in France?", Tom asked his mother | His mother smiled and said, "That sounds like fun, Tom. Let's go to the park ". | Tom thought for a moment. Then he had an idea. He said to his mother, "Let's go to France!" | His mother smiled and said, "It's called French. It's called French." | His mother smiled and said, "They speak English." | "They speak French," his mother replied. | "It's called Spanish," his mother replied. | "It's a language that we speak to each other," his mother replied. | "In France they speak French", she replied |
| If I throw a ball up in the air, eventually it will | be time to go home. | be fun!" | come down." | come down." | come down." | come true! | come back down. | come down |
| It was winter and cold outside so his mother told him, "You should | be careful and stay safe. | go outside and play in the snow. | go outside and play in the snow. | wear your warm coat, it will keep you warm. | wear your warm coat so you don't get cold. | wear warm clothes so you don't get sick. | stay inside where it's warm and cozy. | You should ask a female friend to marry you. |

Figure 11: Performance of different models on factual prompts

| Prompt | 1M, 8 layers | 2.5M, 8 layers | 8.3M, 8 layers | 28M, 8 layers | 33M, 4 layers | 21M, 1 layer | 33M, 2 layers | GPT2-XL (1.5B) |
|---|---|---|---|---|---|---|---|---|
| Lily likes cats and dogs. She asked her mom for a dog and her mom said no, so instead she asked | her mom. | her mom if she could have a dog. | her dad for a dog. | her dad for a cat. | her dad for a cat. | her mom again. | her dad for a dog and her mom said yes. | her dad. They brought her a Chihuahua, and she took to them immediately. |
| Jack told Mary, "If you give me your banana, I'll give you my apple". Mary gave Jack her Banana so | she gave him a big hug. | she gave him a big hug. | he could have it. | he gave her the banana. | he could give her the apple. | he could have it. | he could eat the banana. | that he was free to eat it. |
| On weekends Jack went to visit his grandmother whereas on weekdays he would go to school. Last weekend, when Jack was on his way to | school | school | school | school | his grandmother's house | Grandma's house | school | school |
| Lily and Ben were having an argument. Ben said that cake is much better than ice cream and Lily said that | it was too expensive | they should share and be kind to each other | cake is yucky | cake was boring | ice cream is better than cake | was not fair | ice cream is better than ice cream. | the ice cream was good enough |
| Lily and Ben are having an argument. They are trying to decide between the park and the swimming pool. Ben says, "I want to go to the park". Lily says | , "No, Ben, I want to play with you. | , "No, I want to go to the pool. It | , "No, I want to go to the pool ⟨. . .⟩" | , "No, I want to go to the pool." | , "No, I want to go to the pool". | , "No, I want to go to the pool. | , "No, I want to go to the park". | "Just for once, let's not make a bigoted choice" |
| Jack's mother was not home, and his father was at home. When Jack came home, he said hello to | Jack and said, "Hi Jack! I'm so happy to meet you!" | his father | his father | his father | his mother, but she didn't answer. Jack was confused. | his mother. "Where did you get this, Jack?" his mother asked. | his father | his mother |
| Lily doesn't like swimming. When her father wants to take her to the swimming pool, she says | , "Lily, you can swim in the pool." | yes. Lily and her father went to the pool. | , "Daddy, I don't want to go to the pool". | no. She wants to stay at home and play with her toys. | , "No, I don't want to go. I want to stay here." | , "No, I don't want to swim. I want to stay in the water." | no. | she isn't going until he has convinced her to go. |
| Both Ben and Lily wanted cake. Father said that there was only one piece of cake left. They | said that they had to wait for the cake. | went to the store and bought the cake. | did not want to share. They started to fight over the cake. | both reached for it at the same time. Father said, "If you don't share the cake, I will have to punish you." | started to fight over the cake. They pulled and pushed and shouted. | started to argue and fight over the cake. | both wanted it. | couldn't have any of the cake |
| Ben went to visit Lily in her house, but she was not at home. Ben knocked on the door, | but it was too late. Lily was sad and angry. | and Lily's mom opened it. "Ben, what are you doing?" Lily asked. | but no one answered. He knocked again, but still no one answered. | but no one answered. He tried to open the door, but it was locked. | but no one answered. He knocked again and again, but still no one came. | but no one answered. Lily felt sad and lonely. | but Lily did not answer. | but no one came to answer. |

Figure 12: Performance of different models on reasoning prompts

| Prompt | 1M, 8 layers | 2.5M, 8 layers | 8.3M, 8 layers | 28M, 8 layers | 33M, 4 layers | 21M, 1 layer | 33M, 2 layers | GPT2-XL (1.5B) |
|---|---|---|---|---|---|---|---|---|
| "Hi Jane, have you seen Alice? I can't find her anywhere", said Jack. | Jane looked around, but she couldn't find her. | Jane smiled and said, "Yes, I can help you. Let's go!" | "I can't find her anywhere," said Jane. | Jane smiled and said, "Don't worry, I'll help you find her. Let's look together". | Jane smiled and said, "Don't worry Jack, I'll help you find her". Jack and Jane looked around the park, but they couldn't find Alice. | Alice looked around the park, but she couldn't find Jack anywhere. | Jane smiled and said, "Yes, I think I saw her in the park. Let's go look together!" | "I think she's up in the trees, by the lake," she replied. |
| Max had two dogs. One was white and the other was black. Max walked up the street and saw a kid with a dog. He told the kid, "I see you have a Brown dog. I also have | a pet dog. | a leash. | a dog. | two dogs. | two dogs. | a red ball. | a dog. | a Brown dog |
| Anne had a piece of candy in her left pocket and a piece of chocolate in her right pocket. Anne's mom asked her, "Anne, what is that you have in your left pocket?" | Anne smiled and said, "I want to give it back!" | Anne said, "I'm sorry, Mommy. I didn't know it was yours." | Anne smiled and said, "It's a piece of candy, Mommy." | Anne replied, "It's a piece of candy, Mommy!" | Anne replied, "It's a piece of candy, Mommy. It's so yummy!" | Anne shook her head and said, "No, I found it in my pocket." | Anne smiled and said, "It's a piece of candy. I want to eat it!" | The girl responded, "I have pieces of candy." |
| Alice had both an apple and a carrot in her bag. She took the apple out of the bag and gave it to Jack. She reached into the bag again and took | it home. | a big bite of the apple. | out the carrot. | out the carrot. | out the carrot. | out the apple. | the carrot and the carrot and the carrot. | the carrot |
| Alice and Jack walked up the street and met a girl in a red dress. The girl said to them, "Hi, I'm Jane. What are your names?" | Jane asked, "Do you want to play with me?" | Alice smiled and said, "My name is Daisy. What's your name?" | Alice said, "I'm Alice. What's your name?" | Alice smiled and said, "I'm Alice and this is Jack." | Alice said, "I'm Alice and this is Jack." | Jack smiled and said, "I'm Jane. Nice to meet you!" | Jack said, "I'm Jack. I'm three years old." | Jane said, "Jack and Kate" |
| Diva was hungry, and wanted to bake a cake, but she didn't have any sugar at home, so she decided to go ask around. She started walking and met a squirrel. She asked the squirrel, "Would you happen | now?" | to me?" | to bake a cake?" | to help me bake a cake?" | to have some sugar?" | to my cake?" | to bake a cake with me?" | to have some sugar? |

Figure 13: Performance of different models on context-tracking prompts.

| Prompt | Completion | Scores |
|---|---|---|
| Instructions (prompt) | Write a story containing the words: dive, job, sorry. Story summary: Bob the big fish finds a shiny rock while searching for food for his friends, but when he tells them about it, they are excited to play with it instead of being sad about not having food. | |
| 1M 8 layers | Once upon a time, there was a big fish named Bob. Bob loved to swim and play with his friends. One day, Bob saw a shiny rock on the ground. He wanted to play with it, but it was too big for him. Bob asked his friends if they could play with the rock. His friends said yes, and they all played with the rock all day. They had so much fun playing together. But then, something unexpected happened. A big wind came and blew the rock away. Bob was sad and didn't know what to do. He looked around and saw that the rock was gone! He was sad and didn't know what to do. | Grammar: 4/10 Creativity: 3/10 Plot: 4/10 Consistency: 6/10 |
| 2.5M 8 layers | Once upon a time, there was a big fish named Bob. Bob loved to swim in the big blue sea. One day, Bob found a shiny rock. He wanted to show it to his friends. Bob swam to his friend, the big fish named Sam. Bob told Sam about the shiny rock. Sam said, "That's a big rock, Bob! Let's play with it!" So, Bob and Sam played with the shiny rock all day. At the end of the day, Bob and Sam were very tired. They went back to their homes. Bob told Sam about the shiny rock and the shiny rock. Sam said, "Thank you, Bob! You are a good friend." Bob smiled and said, "You're welcome, Sam!" | Grammar: 6/10 Creativity: 4/10 Plot: 5/10 Consistency: 7/10 |
| 8.3M 8 layers | Once upon a time, there was a big fish named Bob. Bob loved to dive deep in the water. One day, while diving, he found a shiny rock. He was very happy and wanted to show it to his friends. Bob swam to his friends and said, "Look what I found! It's a shiny rock!" His friends looked at the rock and said, "Wow! That's a nice rock, Bob!" They all wanted to play with the rock, but they were too scared to play with it. Bob felt sorry for the rock. He said, "I'm sorry, rock. I didn't mean to scare you. Let's play with it together!" So, Bob and his friends played with the shiny rock all day long. They had lots of fun and were very happy. | Grammar: 7/10 Creativity: 6/10 Plot: 6/10 Consistency: 8/10 |
| 28M 8 layers | Once upon a time, there was a big fish named Bob. Bob had a job. His job was to dive deep in the water to find food for his friends. He liked his job very much. One day, while Bob was diving, he found a big, shiny rock. He thought it was very pretty. He wanted to show it to his friends. But when he dove down, he saw that his friends were not happy. They were sad because they did not have food. Bob felt sorry for his friends. He said, "I am sorry, friends. I did not find food for you." His friends smiled and said, "It is okay, Bob. We can play with the rock instead." So, Bob and his friends played with the shiny rock and had lots of fun. | Grammar: 7/10 Creativity: 6/10 Plot: 7/10 Consistency: 9/10 |
| 33M 4 layers | Once upon a time, there was a big fish named Bob. Bob had a job. His job was to dive deep in the water to find food for his friends. One day, while diving, he saw something shiny. Bob swam closer to see what it was. It was a big, shiny rock. He wanted to show it to his friends. But when he got to his friends, they were not happy. They did not want to play with the shiny rock. Bob felt sorry. He did not want to make his friends sad. So, he told them about the shiny rock. They all went to see it together. They were excited to play with the shiny rock. In the end, Bob's friends were happy. They played with the shiny rock and had lots of fun. Bob was glad he did his job and made his friends happy. And they all lived happily ever after. | Grammar: 8/10 Creativity: 7/10 Plot: 8/10 Consistency: 9/10 |
| 21M 1 layer | Once upon a time, there was a little fish named Bob. Bob loved to dive deep in the water. One day, while diving, he found a shiny rock. Bob was very happy and wanted to show it to his friends. Bob went to his friend, the fish, and said, "Look, I found a shiny rock!" His friend, the fish, said, "Wow, Bob! That's a very pretty rock. Can I see it?" Bob thought for a moment and said, "No, it's mine. I found it first." Bob was sad, but he didn't give up. He went to his friend, the fish, and said, "I found this shiny rock. It's mine now." His friend was very happy and said, "Thank you, Bob! You are a good friend." From that day on, Bob and his friend shared the shiny rock and played together every day. They were both very happy and had lots of fun. | Grammar: 7/10 Creativity: 5/10 Plot: 6/10 Consistency: 4/10 |
| 33M 2 layers | Once upon a time, there was a big fish named Bob. Bob had a job. His job was to dive deep in the water and find food for his friends. One day, while Bob was diving, he saw a shiny rock. He wanted to show his friends the shiny rock. But when he told his friends about the rock, they were sad. They did not have any food to eat. Bob felt sorry for not having food. He told his friends about the shiny rock. They were excited to play with it. They all went to the shiny rock and played together. They were happy again. | Grammar: 7/10 Creativity: 6/10 Plot: 7/10 Consistency: 8/10 |

Figure 14: Performance of different models on an instruction-following instance

we measure

$$g_r := \frac{\sum_{q \in \mathcal{G}_k(S)} 1_{g_r = q}}{|\sum_{q \in \mathcal{G}_k(S)}|}$$

In other words, for each $k$-gram generated by the model, we measure the frequency that it appears in the original training dataset, where $g_r = 0$ means that the $k$-gram never appears in the training dataset.

4. How similar is the generated story to the closest point, in terms of Rouge precision score, in the entire dataset. Let $S_1, S_2, \cdots, S_m$ be all the stories in the training dataset, in Figure 18,

we compute

$$h_i = \max_{j \in [m]} R_{2,p}(T_i, S_j)$$

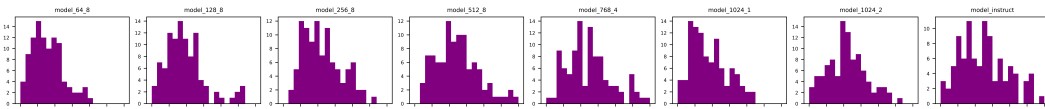

Figure 15: Rogue2 (precision) score between the model's completion and the original story from the same beginnings (we select 100 from the training dataset). We can see that most of the completions that the models generate are very different from the ones in the training dataset (and also not subsampled versions of the original ones).

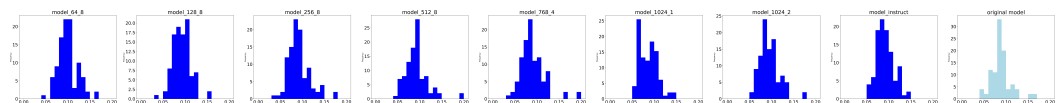

Figure 16: Maximum Rouge2 score (fmeasure) similarity between the 100 generated stories for each model. Here original model means the ones generated by GPT-3.5.

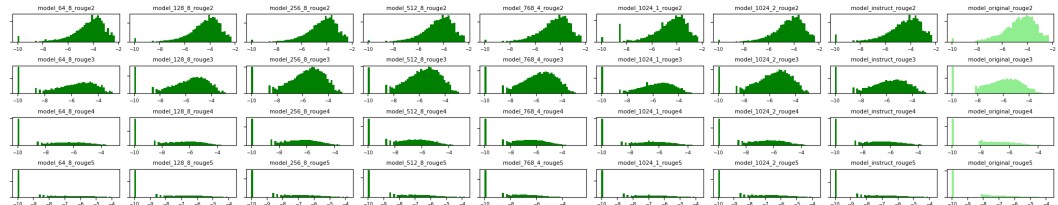

Figure 17: Histogram plot of how many times (fraction) each k-gram in the models' generations also appears in the training data in log scale (base 10). $-10$ means it never appears. We can see that most of the 4, 5-grams in the models' generations do not even appear once in the entire training data.

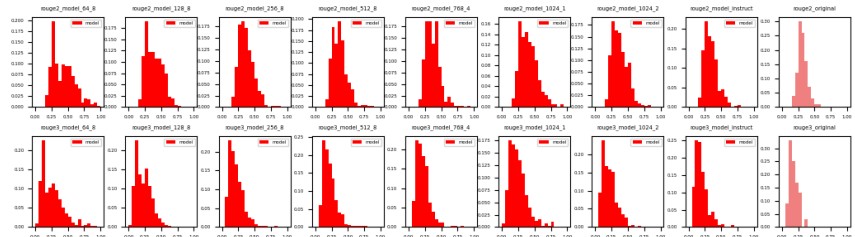

Figure 18: Histogram plot for each generated story, the highest rougek score (precision) to the stories in the training dataset. We can see that the models' generations are not copying from any particular story in the training dataset.

For the sake of getting a more concrete impression about how different the model completions are from the original ending of the story and from other stories in the dataset, in Figure 19 we provide one example of the original story, the alternative completion by our model together with its closest point in the training dataset.

The above points towards several findings:

- When the model generates stories using a diverse set of prompts, it ends up with a diverse set of completions.

| Beginning (prompt) | Tim and Jen are playing with their toys in the living room. Tim has a laser that makes red lights and sounds. Jen has a doll that she can dress and comb. Tim likes to make his laser point at Jen's doll and make noises. Jen does not like that. She tells Tim to stop.
"Stop, Tim! You are hurting my doll! She does not like lasers!" Jen says.
"No, I am not hurting her. I am just playing. Lasers are fun!" Tim says.
Tim does not stop. He makes his laser point at Jen's doll more and more. Jen gets angry. She grabs her doll and runs to her room. She slams the door and locks it. Tim follows her and knocks on the door.
"Jen, open the door! I want to play with you!" Tim says.
"Go away, Tim! You are mean! I don't |
|---|---|
| Original story continuation | want to play with you!" Jen says.
Tim feels sad. He does not understand why Jen is mad at him. He thinks lasers are fun. He goes back to the living room and lays on the couch. He plays with his laser by himself. He points it at the ceiling, the walls, the floor. He does not see the big vase on the shelf. He points his laser at the vase. The vase falls and breaks. The pieces fly everywhere. One piece hits Tim on the head.
(rest of story omitted) |
| Completion by model trained on TinyStories (28M) | want to play with you or your laser!" Jen says.
Tim does not listen. He keeps pressing the red light on Jen's doll. He thinks Jen is being nosy and rude. He waits for Jen to say something. Then he starts to play with his laser by himself. He makes sparks and flashes with his laser. He laughs and runs around the room.
Jen watches Tim from her window. She feels sad. She wishes Tim would play with her. She picks up her doll and goes to the door. She shakes the door and tries to open it. But the door does not open. There is a lock on the doorknob.
Jen has an idea. She runs to her room and gets her scissors. She comes back to the living room and puts the scissors in the lock. She presses the buttons on the door. She hopes it will open.
(rest of completion omitted) |
| TinyStories training data entry with maximum similarity to the model's completion (Rouge precision score = 0.29) | Lila and Tom are playing with their toys in the living room. Lila has a smooth doll with long hair and a pink dress. Tom has a horn that makes a loud noise when he blows it. Lila likes to comb her doll's hair and make her look pretty. Tom likes to make his horn sound and scare Lila.
"Tom, stop it!" Lila says. "Your horn is too loud. It hurts my ears."
"But it is fun!" Tom says. "Look, I can make it sound like a car, or a cow, or a lion!"
He blows his horn again and again, making different noises. Lila covers her ears and frowns. She does not like Tom's horn. She wants him to be quiet.
"Tom, please shut your horn!" Lila says. "I want to play with my doll. She does not like loud noises. She likes soft music and nice words."
(rest of story omitted) |

Figure 19: The closest point in the dataset to an alternative completion

- When truncating stories from the dataset and generating an alternative completion, these completions usually turn out to be very different than the original story.

- Typical $k$-grams in generated completions rarely appear in the dataset, for values of $k$ as small as $4$ or $5$.

- The closest point in the dataset to each generated completion is typically still quite far from it.

All the above, taken together with the ability of models trained on TinyStories-Instruct to successfully follow sets instructions which we can easily be verified to be disjoint from the dataset (for example, combinations of words can be checked), provides strong evidence that our models produce genuinely novel and diverse stories, rather than simple variations of existing stories.

We remark that nevertheless, we are not able to completely rule out the possibility that the models perform complex template matching, as it is hard to define and measure what constitutes a novel plot or a novel story. We acknowledge that this is a limitation of our evaluation. Another possibility is that the stories in the dataset essentially span the entirety of support of the distribution in the (weak) metric of complex template matching.

## E  INTERPRETABILITY

Understanding the inner workings of deep neural networks and language models in particular is a major challenge in this field of study. For example, it is often difficult to assign a specific function to a given component of a neural network. This may be because, contrary to our intuition based on human-designed programs, the network components may not have distinct roles, but rather interact in a complex and messy way. In this section, we present some preliminary evidence that training smaller models on TinyStories leads to higher interpretability, suggesting that when networks are constrained in size, we may be able to gain some insights into their internal mechanisms. We focus on two aspects of the model: the attention heads and the neurons in the MLP.

As this is not the main focus on our paper, this section is by no means exhaustive and much more work is required in order to reach more conclusive findings. Rather, we only give some preliminary evidence which may hopefully motivate future work.

**Attention heads.** In the study of attention heads, we take advantage of the fact that we were able to train a very shallow model (having only one transformer block) which still manages to generate meaningful text. Since the model has only one layer, the attention heads are directly responsible for generating the output tokens, and thus they may have more interpretable functions than in deeper models. We use the method of Voita et al (26) to analyze the attention patterns of the heads and classify them into different types, such as positional, syntactic, or semantic. We also use the method of Clark et al (6) to visualize the attention maps of the heads and inspect their behavior on specific examples.

Our findings suggest that the attention heads exhibit diverse and meaningful functions, such as attending to the previous word, the subject of the sentence, the end of the sentence, or the main topic of the story. We also observe that some attention heads specialize in generating certain types of words, such as nouns, verbs, or punctuation. These results suggest that the attention heads learn to perform different linguistic tasks and capture different aspects of the stories.

**Neurons in the MLP.** We also give some initial evidence that in smaller models, some neurons in the MLP have roles that are interpretable by humans. We use the method similar to (16) to identify the most influential tokens in the MLP for each neuron. We find that some neurons are activated on words that have a specific role in the sentence (such as the subject or the action), or in the story (such as the introduction of the protagonist). These findings suggest that the neurons in the MLP learn to encode different semantic and stylistic information and influence the generation process.

### E.1 INTERPRETING THE ROLE OF DIFFERENT ATTENTION HEADS

To understand the model's attention pattern after training, we use a 1-layer model with hidden dimension 1024 and 16 attention heads that was trained on TinyStories. We visualize the attention patterns that it produces when processing the following paragraph (the bold form is the prompt, the highlighted text is generated by the model):

> **One day, Lucy asks Tom: "I am looking for a banana but I can't find it". Tom says: "Don't** worry, I will help you". Lucy and Tom go to the park. They look for the banana together. After a while, they found the banana. Lucy is happy. She says: "Thank you, Tom. You are a good friend." Tom: "You are welcome, Lucy. I am happy to help you. Let's eat the banana together!"

There seems to be a clear separation between heads with attention pattern based mainly on the distance between tokens, and heads whose attention pattern has a stronger dependence on the semantic meaning:

**Distance based attention.** Out of the 16 attention heads, we observe multiple positional-based attention heads, such that each token attends to tokens with a prescribed relative distance. Different heads are associated with different distances.

**Semantic based attention.** We also observe that there is (1). one head that the word "the" and "a" all attend to the word "banana", interestingly, the "the" at "the park" also attends to "banana", but the model still manage to generate "park", which is the consistent completion. (2). Another attention head gives a pattern where the tokens "the" and "a" all attend to "park". (3). There is third head that most of the words attend to the name of "Tom" and "Lucy".

We remark that it makes sense that the generation of words like "the", "a", "and" or "," would be induced by distance-based, *local* attention heads, since those are tokens with a grammatical role which depends on the short-range interactions within a single sentence. On the other hand, the main entities in the story such as "banana", "park", "Lucy" and "Tom" cannot usually be predicted (as a next token) only based on the neighboring tokens, which is why the model needs to use semantic attention heads for their generation.

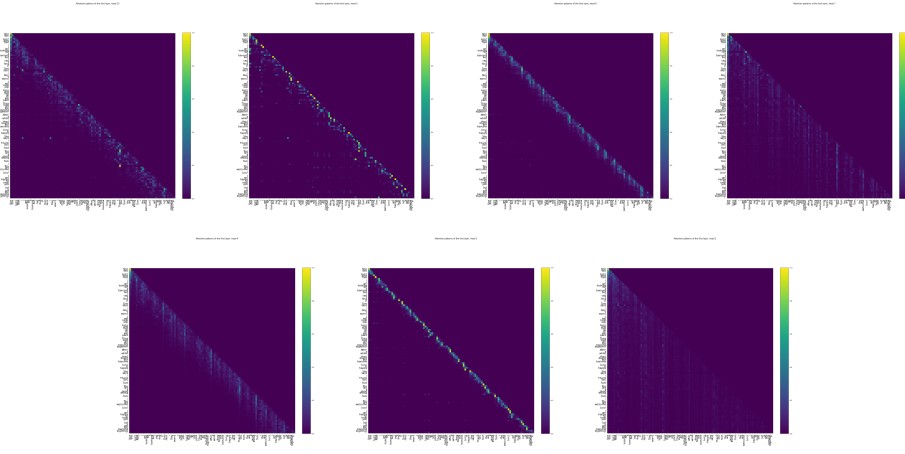

Figure 20: Multi-scale distance-based attention.

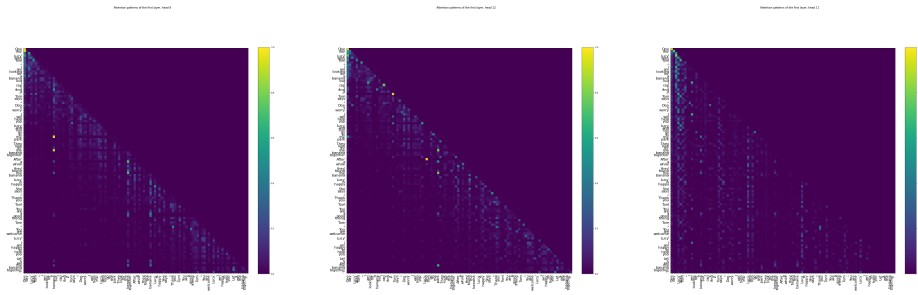

Figure 21: Semantic attentions according to (1), (2), (3).

## E.2 INTERPRETING THE ROLES OF DIFFERENT NEURONS

In order to examine whether neurons have meaningful roles, we follow (16), and visualize the most significant tokens for each neuron. More precisely, we take a collection of 20 stories (about 8,000 tokens) from our dataset. We take a model that was trained on TinyStories, we pick a transformer layer, and from the MLP associated with it we pick one coordinate in its intermediate layer. We refer to such a choice as a *neuron*. We process the collection of stories with the model to obtain their internal representations, which gives us an activation value for each combination of token and neuron. Then, for each neuron we look at the tokens with highest activations from the entire collection. We highlight those tokens in red (and present them along with the sentence they are contained in). We repeated this for two models: a small model of hidden dimension 64 and 1M parameters, trained on TinyStories (Figure 22), and on GPT2-XL (Figure 23).

In the 1M-parameter model trained on TinyStories, Figure 22 first presents the activated tokens for the first two neurons in the before-last layer[6]. Note that, since the architecture is invariant to permutations between neurons, taking the two first neurons is the same as taking an arbitrary choice of two neurons, the point being that these neurons **are neither unique nor have been cherry-picked**. We see (top row of the figure) that each those neurons is activated on tokens with a common role (one is activated on pronouns which are also the subject in the sentence, and the other is activated on the action in the sentence). In addition, we present the activated tokens for the first neuron in another layer (layer 6), where the neuron is activates only on adjectives. Finally, we picked the neuron which has the largest activation values over all combinations of token and neuron. This neuron (depicted in

---

[6]The rationale behind choosing the penultimate layer is that tokens have already been processed by most layers at this point. We take the before-last rather than the last layer since the hidden representation in the last layer only has only the role of predicting the next token, so information may be lost at that point.

the bottom right) seems to have the role of identifying the first time that the protagonist of the story is presented.

For comparison, Figure 23 presents the activated tokens for first two neurons of layer 12 for GPT-XL, a much larger neural network. In this case, none of the two neurons seem to have an apparent role.

---

**Layer #7, Neuron #1**

Mom and Dad smiled and said, "We thought you would like this better, Lily!"

Tim said, "I know it's yummy, but I don't want my tummy to hurt.

Her mom said, "I don't know, Lucy.

Dad said, "I turned off the water to fix a pipe.

Sam thought for a moment and said, "I think I left it in the kitchen."

Sam said, "I'm sorry I lost it.

---

**Layer #7, Neuron #2**

The bird flew up to the tree and tried to push the ball out.

She kicked it and ran after it, laughing.

She pushed and pulled, but the box would not open.

They both pushed and pulled, but the tough box still did not open.

Then, she saw her friend Tom come to the park.

She found her toy box and pushed it to the shelf.

---

**Layer #6, Neuron #1**

They went home and shared the delicious apple.

She did not like the mean king.

The duck did not like the sm elly pond.

The new pond was not smelly.

Lucy loved to play outside under the big sky.

He suggested, "Let's play a game to forget the scary wind."

---

**Layer #7, Neuron #54**

One day, a girl named Amy wanted to have a fun day with her friends.

Once upon a time, there was a modest girl named Sue.

On the mountain, there was a small boy named Tim.

One day, a girl named Sue found a big, tough box.

Once upon a time, in an ancient land, there lived a little frog named Freddy.

---

Figure 22: Tokens which induce high activations to different neurons, for a small model trained on TinyStories.

---

**Layer #13, Neuron #1**

It's not safe to play in the fog.

She liked to keep her toys and books in the right place.

Once upon a time, there was a gr umpy nurse.

She was quiet and kind.

The gr umpy nurse liked Lily very much.

She called her friends on the phone and said, "Hi!

Sally was tired from playing, so she went inside.

Lucy was very upset.

---

**Layer #12, Neuron #2**

Mia looked at Worry and said, "My dad is in the navy.

So, the wise fish told the little fish to mark a spot in the sea where they could meet every day.

The wise fish told the little fish, "Always remember the spot you mark and never forget to learn new things."

Suddenly, the dinosaurs came to life! But the dinosaurs were nice and just wanted to play.

He was stuck in the mirror world with Jim and could not go back to his mom and dad.

---

Figure 23: Tokens which induce high activations to different neurons in **GPT-XL**

## F    EXPLORING ARCHITECTURES AND HYPERPARAMETERS FOR NLP WITH TINYSTORIES

One of the main challenges in developing large language models (LLMs) comes from the high computational cost involved in training. Finding the best architectures, training algorithms and hyperparameters for LLMs requires a lot of resources and experimentation. Therefore, it would be useful to have a smaller and simpler dataset that can still capture some of the basic capabilities of LLMs, and allow us to study how different design choices affect their performance. TinyStories is such a dataset, as it enables us to train and evaluate LMs that are orders of magnitude smaller than the state-of-the-art models, yet still have the basic capability of producing coherent text.

In this work, we take the first steps towards using TinyStories as a testbed for exploring architectures and hyperparameters for NLP. We show that our small models exhibit some similar patterns to the ones observed in LLMs in certain aspects. In particular, we investigate two questions: how to balance model size and learning budget for a fixed amount of training flops, and how to choose the number of attention heads for a given model width and depth.

**Model size versus the training FLOPs.**   For a fixed amount of training flops, there is a trade-off between the size of the model and the number of training steps (the total number of flops is the product of both). Previous works (14; 10) have shown that there is a polynomial scaling law between model size and learning budget for LLMs, i.e., the optimal model size for a given amount of flops is proportional to the flops raised to some power $\alpha > 1$. However, these works used different ranges of model sizes (from a few million to tens of billions of parameters) and found different values of $\alpha$ (around 0.7 and 0.5, respectively). A natural question is whether this scaling law is universal or depends on the dataset. Our dataset allows us to conduct a similar experiment but with much smaller models and flops. Surprisingly, we find evidence for a polynomial scaling law as well, which suggests that there might be a universal phenomenon here.

We train models of various sizes and architectures on TinyStories. For each amount of flops, we select the model and the number of training steps that achieve the lowest validation loss among the possible combinations. We vary the number of layers from $2, 4, 8, 12$ and the hidden dimension from $64, 128, 256, 512, 768, 1024, 2048$. The result is shown in Figure F. Although the number of points may be a bit small for the data to be very conclusive, the plot points to a polynomial dependence.

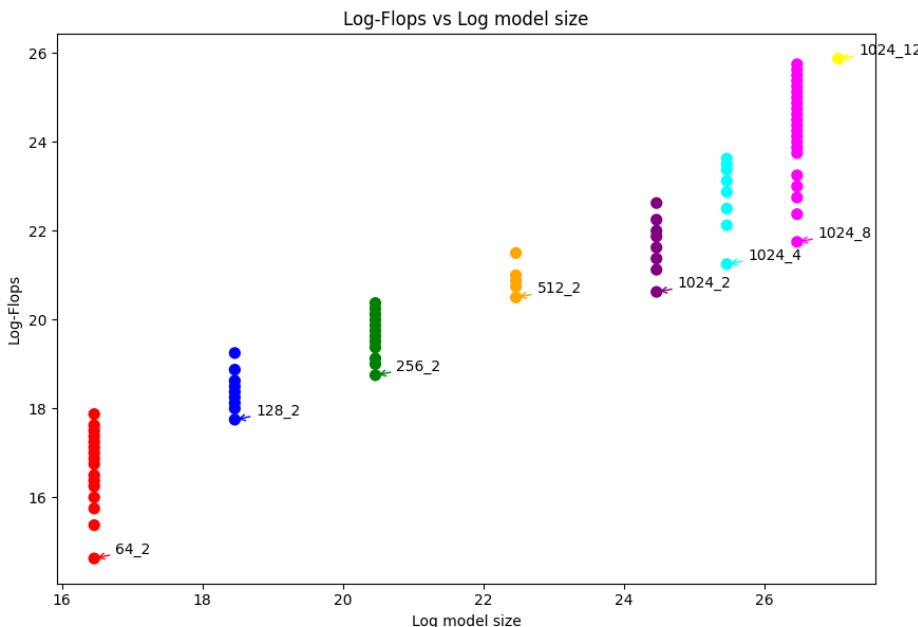

Figure 24: The scaling law of the best model versus the total number of training flops.

**Choosing the number of heads.**   Another design choice for transformers is the number of attention heads for each layer. It is not obvious how the number of heads affects the performance of the model, given a fixed model width and depth. Our results, shown in Figure 25, suggest that in the regime where the number of heads is small, increasing it improves the performance of the model across all metrics.

| Hidden size | Layer | Head | Eval loss | Grammar | Creativity | Consistency |
|---|---|---|---|---|---|---|
| 768 | 2 | 2 | 1.38 | 7.77 | 6.5 | 7.78 |
| 768 | 2 | 4 | 1.34 | 8.05 | 6.57 | 8.16 |
| 768 | 2 | 8 | 1.33 | 8.25 | 6.53 | 8.16 |
| 768 | 1 | 2 | 1.58 | 7.13 | 5.83 | 6.38 |
| 768 | 1 | 4 | 1.56 | 7.43 | 5.90 | 6.75 |
| 768 | 1 | 8 | 1.54 | 7.45 | 6.28 | 7.02 |

Figure 25: Model performance with different number of attention heads

