# OpenReview forum: "TinyStories: How Small Can Language Models Be and Still Speak Coherent English"
_ICLR.cc/2024/Conference — ICLR 2024 Conference Withdrawn Submission_

### Official Review · Reviewer_5ySx · 2023-10-29

**Soundness:** 1 poor
**Presentation:** 2 fair
**Contribution:** 2 fair
**Rating:** 1
**Confidence:** 4

**Summary:**

This paper introduces a dataset, namely TinyStories, which was produced by ChatGPT as well as GPT-4, and which contains only simple sentences. The author argued that with this simple dataset simpler model can produce fluent and consistent stories.

**Strengths:**

- The subject matter of the present paper looks very interesting, namely the relationships between language complexity, the size of LM, and the quality of the text generated by the LM. My impression is that there is an impossible trinity between them.
- The TinyStories dataset would be very useful for the community.

**Weaknesses:**

In the very first place, the general research question of this paper is very unclear to me. Given the title, the paper seems to focus on exploring how small a model can be but can still produce fluent texts. But, after I started to read the paper, I found this was not the case. Both the research question and the response were problematic. I am saying this based on the following reasons:
-  The research question is ill-defined. If the title is the main research question that this study intended to answer, then it is hard to know what does "coherent" actually mean in the context of this paper. This is unclear because, later on, the paper (page 1) says Tinystories was designed to capture the ability of grammar, vocabulary, facts, and reasoning. For me, none of this is about coherent.
- Another inconsistency is that by the end of the first paragraph, the paper says the question that this paper focuses on is "Is it possible to design a dataset that preserves the essence of natural language, while reducing its breadth and diversity" (which is clearly very different from what the title says). This is again not about coherence. It is also questionable why "diversity" is not a kind of "essence of natural language". Also, interestingly, on the same page, the paper says that one of the goals of TinyStories is to train a model that can produce *diverse* stories.
- The design of the whole study looks very unscientific to me. Regardless of what exactly the research question is, this study has too many free variables, at least including, the complexity of the language in the dataset, the size of the dataset and the size of the model. In this sense, I don't think the conclusions are reliable and generalisable.

Apparently, a clear trade-off exists in the dataset, namely the complexity of the language. (btw. for me, this is a type of "essence of natural language".) I don't think it is a limitation of the dataset but do think it is an important aspect to research and discuss. Nonetheless, unfortunately, this paper seems to totally overlook this important aspect.

Given the task defined, this paper also proposed an evaluation framework, the reliability and rationality of which are both questioned: (1) it is quite unreasonable to train a model using contents produced by GPT and, meanwhile, also evaluate the same model using the same GPT. (2) only 50 prompts are used in the evaluation. Since only automatic evaluation was used in this study, I wonder why more prompts were not considered. (3) it is also unclear why only assessing grammar, creativity and consistency. None of them is about either coherence or fluency. More importantly, creativity seems to be irrelevant to any goals mentioned in the introduction.

Generally speaking, for me, the paper looks more like a position paper rather than a regular paper for ICLR as while no serious evaluation had been conducted, meanwhile, the last 4 pages are all about example outputs.

**Questions:**

See my comments above.

---

### Official Review · Reviewer_itNL · 2023-10-31

**Soundness:** 1 poor
**Presentation:** 2 fair
**Contribution:** 1 poor
**Rating:** 3
**Confidence:** 3

**Summary:**

This paper introduces a synthetic dataset of short stories generated using prompting on two commercial LLMs. The dataset is then used to train a small LM. The motivation is to "build much smaller models that produce fluent and consistent stories with several paragraphs that are diverse and have almost perfect grammar, and demonstrate reasoning abilities". The small LMs are then evaluated using a commercial LLM and compared to four baselines.

**Strengths:**

The goal of producing powerful yet smaller LMs is an interesting one. Focusing on distilling a model which contains core linguistic competence is also attractive. The idea of focusing the distillation on simple language for young children is novel as far as I can tell.The goal of producing powerful yet smaller LMs is an interesting one. Focusing on distilling a model which contains core linguistic competence is also atractive.

**Weaknesses:**

From the motivation it is not clear whether the main goal is to generate small generic models or models for younger readers (3 to 4-year-olds). It is not clear which could be the use of the generated model, apart from the fact that it is smaller.

The concepts of "essence of natural language", "breadth", "core elements of natural language" are not defined. A proper definition and evaluation would make the proposal stronger.

The evaluation is fully automatic, with only a handful of examples hand-labeled by the authors. As the paper proposes to use GPT-4 for evaluation, there should be an specific experiment to validate whether GPT-4 is effective when evaluating creativity, grammar etc. of models. In particular, the paper compares a model distilled from one of the GPTs (via the synthetic corpus), and it would be important that the evaluation scores produced by GPT-4 are not biased towards preferring models distilled from GPT-3.5 and GPT-4.

The motivation mentions the goal of producing text which is understandable by 3 to 4-year-olds, but no evaluation is done about whether this is effectively the case.

On the technical side, this paper is about distillation, so in that sense it does not compare to other distillation techniques, which could provide better results. The proposed baselines are models built from scratch.

**Questions:**

When comparing language models, it is also important to mention the size of the training corpus. Could you mention the amount of tokens used by each of your models?

---

### Official Review · Reviewer_tD6p · 2023-11-02

**Soundness:** 2 fair
**Presentation:** 3 good
**Contribution:** 2 fair
**Rating:** 3
**Confidence:** 4

**Summary:**

This paper presents a new dataset and proposes a new paradigm for language model evaluation. The dataset consists of stories that are written in a simplified manner. Explicitly, it is intended that these stories consist of only the vocabulary of a 3-4 year-old child, make use of a narrower set of grammatical structures, and employ a limited subset of world knowledge. The authors state that the motivation behind this is to test whether the ability to generate coherent text requires large models and complex architectures (I would humbly suggest that the authors instead focus more on this research question than on the release of the dataset). They then propose an evaluation protocol for smaller language models, which involves querying GPT-4 for quantitative measures of the quality of text generated by these models. The paper then presents several analyses of models trained on their proposed dataset. They analyze how smaller models’ abilities to generate creative, coherent and grammatical text changes as a function of various hyperparameter settings.

**Strengths:**

* The new dataset covers an interesting sub-domain of NLP, and has a variety of potential uses
* The paper gives some interesting findings about how model capabilities relate to their size, including observations about grammaticality and consistency

**Weaknesses:**

* While the motivation behind the dataset has some nice intuition, there are no concrete assessments of the presence of the properties that the authors claim the dataset has.  Concretely, there are no assessments of grammaticality, factual content, vocabulary coverage, etc. and how these quantities compare to larger, human-generated corpora.
* On a similar note, there does not seem to have been any validation of the dataset, which was machine-generated.
* There are many claims about children’s language usage that are either not supported, or are not elaborated on to a degree necessary to believe the claims made by the authors. For example, what are the grammatical structures and facts used/known by a child? What checks are done to ensure that these constraints are adhered to?
* Some of the experiments seem quite ad-hoc. For example, the results of figure 6 are color-coded  “according to their success (green), failure (red), or partial success (yellow),” but the criterion for success is never defined, nor is the evaluation setup provided.
* In general, text in the figures is very difficult to read

**Questions:**

* Why should evaluation for smaller models be different than that of larger models? Ultimately, do we not care about the same criteria?
* When you say “spanning the knowledge base of a 3-4 year old child”, to me this implies factual knowledge
* Are there quantitative measures of the diversity, in comparison to other (human-generated) natural language corpora
* How are the generated texts checked for quality?
* In 4.3, it is claimed that “the closest point in the dataset to each generated completion is typically still quite far.” How is this practically evaluated?
* On page 3, what is meant by “more restricted in terms of its content”

---

### Official Review · Reviewer_YfoG · 2023-11-10

**Soundness:** 2 fair
**Presentation:** 2 fair
**Contribution:** 2 fair
**Rating:** 5
**Confidence:** 3

**Summary:**

This paper is a sum of contributions around two ideas: i) ask to GPT to generate a simple dataset (with words that a typical 3 to 4-year-old child can understand) and ii) use GPT4 to score LLM.

**Strengths:**

One goal is to build a synthetic dataset with a "simple" language to better understand how language is learnt by LLM. The idea is interesting and some hints provide interesting assumptions, like : model depth is more important for consistency than for grammar,  the grammar can be quickly learnt, ...

**Weaknesses:**

This paper mostly relies on GPT4 (or 3.5): both for generation and evaluation.
The small models are never clearly described (their architectures, the exact training process, ... ).  This is however important. While many examples are provided in the paper, it is difficult for the reader to find out which model produces them. Moreover, how do you generate the texts with SLM ? This is an important question, especially when the creativity is evaluated. Can you claim that you really evaluate the creativity or the decoding strategy.  It looks like some part of the paper are bit naive and corresponds more to a fancy use of GPT 4 than a real scientific study.

**Questions:**

Why the interpretability section is considered as a contribution in the introduction, while it is  only in appendices ?